# Modulatory action of *Bryonia alba* on the immune system in cyclophosphamide induced immunosuppression in BALB/c mice

**Vara Prasad Saka**[1], **Godlaveti Vijay Narasimha Kumar**[1]*, **Abanti Goswami**[1], **Bharat Kumar Reddy Sanapalli**[2], **Pankaj Gupta**[3], **Digvijay Verma**[4], **Subhash Kaushik**[4]

**1** Department of Pharmacology, Drug Standardization, Dr Anjali Chatterji Regional Research Institute for Homoeopathy, Kolkata, West Bengal, India, **2** Department of Pharmacology, School of Pharmacy and Technology Management, SVKM's Narsee Monjee Institute of Management Studies (NMIMS) Deemed-to-be University, Jadcherla, Hyderabad, India, **3** Department of Pharmacology, Drug Standardization, Dr. D P Rastogi Central Research Institute for Homoeopathy, Noida, Uttar Pradesh, India, **4** Drug Standardization, Central Council for Research in Homoeopathy, New Delhi, India

* narasimha.spsp@gmail.com

## Abstract

Oxidative stress and inflammation are the most common pathologies in immune-compromised diseases and cancer treatments. The study examined the immune stimulation properties of *Bryonia alba* (BA) in different potencies (6C, 30C, and 200C) on a BALB/c mice model with a compromised immune system induced by cyclophosphamide (CPM) at a dose of 80 mg/kg. Seventy mice (35 males and 35 females) were randomly distributed into seven groups of 5 animals/sex. Mice treated with different potencies of BA showed notable improvements in various immune parameters, including RBC, WBC, and Hb levels, as well as thymus and phagocytic indices. Treatment also increased serum levels and splenic mRNA expression of IL-2, IL-4, TNF-α, and IFN-γ. The histopathology analysis showed that the spleen sections of the normal group exhibited intact white and red pulp. In contrast, the sections of the CPM group exhibited disrupted and atrophied white pulp. The treatment with BA maintained the spleen in a preserved state, with the white and red pulp remaining intact and a higher density of lymphocytes. The results indicate that BA could serve as a valued immunostimulant agent when administered with chemo-therapy. However, additional research is required to assess the immunostimulatory effects of BA in immune-compromised infections.

## Introduction

The human immune system is a robust defensive system that provides protection against pathogens and foreign antigens [1, 2]. The immune response encompasses both innate and adaptive immunity, which relies on vital immunological organs like the thymus and spleen, along with immune cells that are essential for maintaining the body's immunity. Nevertheless, the immune system's defense activity can be affected by various factors such as age, metabolic

**Data Availability Statement:** All relevant data are within the manuscript and its Supporting Information files.

**Funding:** Central Council for Research in Homoeopathy, India sponsored the study under the Intramural Research Scheme (4-13/2019-20/CCRH/Tech/DS/Kolkata/DSU/3255). The funders had no role in study design, data collection and analysis, decision to publish, or preparation of the manuscript.

**Competing interests:** The authors have declared that no competing interests exist.

disorders, stress-related conditions, and immune-related infections [3–5]. Immunosuppression refers to a state of either temporary or permanent weakened immune system function, resulting from damage to the immune response. This condition leads to a reduced ability to combat diseases and a compromised immune system [6]. In recent times, it has been noted that a weakened immune system is a common symptom of a range of illnesses, including COVID-19 caused by SARS-CoV-2 [7]. In addition to that, the rise of infectious diseases such as HIV, Ebola hemorrhagic fever, SARC, Zika virus-related encephalopathy, and others, has sparked interest in studying immune-related infections [8].

Augmentation of the immune response is desirable to prevent infection in states of immune suppression, fight established infections, and with other diseased states. Since ancient times, millions worldwide have been using traditional medicinal systems to develop/boost their immunity [9]. Homoeopathic is one such ancient popular Ayush medicinal system that has been used by millions of people around the world. When administered based on symptom similarity, homoeopathic medicines might alter the state of immune homeostasis and stimulate the body's defense mechanisms to combat the pathogen or correct any immunodeficiency or hypersensitivity [10]. *Bryonia alba* (BA), a member of the Cucurbitaceae family, has been traditionally used in Homoeopathic and herbal medicine for a wide range of conditions including frontal pain, cough, peritonitis, serous tissue inflammation, typhoid, pneumonia, jaundice, rheumatism, heart health issues, and brain conditions characterized by serous exudation and facial neuralgia [11, 12]. Its roots are commonly employed in the preparation of homoeopathic mother tinctures i.e. ethanolic solutions and have been noted for their antirheumatic and antiphlogistic effects [12]. Historically, BA has been utilized during the Spanish flu in 1920 [13–15] and as a genus epidemicus for chikungunya prevention, demonstrating a risk reduction of at least 20% in developing the illness, which supports its use in treating chikungunya fever and chronic post-chikungunya arthritis. Recent studies have also described the efficacy of BA homoeopathic therapy against COVID-19 [16, 17] pointing to its immunomodulatory and antioxidant properties.

Various studies have employed multiple methodologies to elucidate its antioxidant capacity and assess its potential toxicity. Irina et al. (2019) conducted a comprehensive study on the antioxidant properties of BA utilizing a range of assays. Their study employed the DPPH (2,2-diphenyl-1-picrylhydrazyl) assay, CUPRAC (cupric reducing antioxidant capacity) assay, FRAP (ferric reducing ability of plasma) assay, TEAC (Trolox equivalent antioxidant capacity) assay, EPR (electron paramagnetic resonance) method, and SNPAC (silver nanoparticles antioxidant capacity) assay. These diverse methods provided a robust evaluation of the antioxidant capabilities of BA. The findings indicated significant antioxidant activity in the aerial parts of BA highlighting its potential as a promising antioxidant agent [11]. In another significant study, Ielciu et al. (2019) explored the antioxidant properties of BA through various biochemical assays, including horseradish peroxidase and myeloperoxidase assays. This research also incorporated cellular antioxidant tests to assess the inhibitory effects on reactive oxygen species (ROS) produced by neutrophils isolated from equine blood and human monocyte-derived macrophages in vitro. The results demonstrated that BA significantly inhibited peroxidase-catalyzed reactions on substrates such as Amplex Red and L012, showcasing its potential in mitigating oxidative stress [18]. Both Irina et al. (2019) and Ielciu et al. (2019) extended their research to assess the toxicity of BA. Irina et al. (2019) evaluated whole organism toxicity using zebrafish larvae and performed anti-plasmodial tests on two *Plasmodium falciparum* strains. The results revealed an absence of toxicity, underscoring the safety profile of BA. Similarly, Ielciu et al. (2019) confirmed the non-toxic nature of BA through cellular models and zebrafish larvae, reinforcing its safety for medicinal applications.

Additionally, BA has been shown to be an effective anti-inflammatory agent, as demonstrated by Mert et al. (2019). They investigated the anti-inflammatory, antinociceptive, and antioxidant properties of BA. To assess the anti-inflammatory effects, they used the carrageenan-induced hind paw edema model and the Whittle method in mice. The antinociceptive activities were evaluated using the p-benzoquinone-induced abdominal constriction test and the tail flick test in mice. The study also examined the antioxidant activities through DPPH-radical-scavenging, ABTS radical–scavenging, total antioxidant activity, and hydroxyl radical–scavenging assays. Their findings revealed that the extract derived from BA roots exhibited significant anti-inflammatory, antinociceptive, and antioxidant activities, indicating its potential therapeutic benefits [12].

BA has shown potential in treating SARS-CoV-2 infections, as demonstrated by Goswami et al. (2022). They investigated the antioxidant properties of BA through the inhibition of heme oxygenase-1 (HO-1) and assessed its therapeutic potential against SARS-CoV-2. In this study, 13-day-old embryonated *Gallus gallus domesticus* eggs were used, which were induced with a spike protein receptor binding domain to create a cytokine imbalance. The administration of BA 200C resulted in a significant increase in IL-10 levels and a decrease in other cytokines following an antigen challenge. These results suggest that BA may possess antiviral properties against the pathogenesis of SARS-CoV-2, highlighting its potential as a therapeutic agent for COVID-19 [19].

The success of BA in treating viral infections highlights its immunomodulatory and antioxidant properties, indicating that BA may also boost immune function in immunosuppressed conditions, i.e., reduction in the efficiency of the immune system's ability to fight infections and diseases. However, BA's immunostimulant properties in immunosuppressed situations have not yet been explored. Therefore, the current study aimed to examine the immunostimulant effects of different homoeopathic potencies (refers to the levels of dilution and succussion of a substance in remedy preparation. These are represented by scales such as centesimal (C), decimal (X), or millesimal (M). Higher potencies involve greater dilution and are thought to increase the remedy's therapeutic effectiveness in Homoeopathy) of BA in immunocompromised BALB/c mice.

In homeopathy, potentization is a fundamental process wherein mother tinctures or ethanolic solutions derived from crude drugs are progressively diluted and subjected to vigorous shaking and forceful striking, known as succussion, to produce potentized dilutions or potencies [20]. Most homeopathic medicines are designated in the centesimal (C) scale, where each stage involves dilution by a factor of 100. A 6C potency indicates that this process has been repeated six times, resulting in a dilution of 1 in $10^6$, while a 30C potency means the process has been performed thirty times, yielding a dilution of 1 in $10^{30}$ [21].

## Materials and methods

### Reagents & chemicals

*Bryonia alba* Potencies (6C, 30C, and 200C) and Dispensing alcohol (90%) were procured from Hahnemann Publishing Co. Private Ltd, India. Cyclophosphamide (CPM) (Sigma, USA), Levamisole (Sigma, USA), Sodium carbonate (Sisco Research Laboratories Pvt. Ltd., India), Indian liquid ink (Parker, USA) hematoxylin (Sisco Research Laboratories Pvt. Ltd., India) and eosin (Hi-Media, India), and all other consumables from reputed local vendors.

### Selection, randomization, and blinding of animals

The animals and experimental procedures under this study were approved by the Institutional Animal Ethics Committee of Dr. Anjali Chatterji Regional Research Institute for

Homoeopathy, with proposal no. DACRRIH/CPCSEA/IAEC/ 2021/005. During the study, animals were housed, handled, and cared for in compliance with the guidelines for Laboratory Animal Facility issued by the CCSEA, Govt of India. In the study, Isoflurane (Raman &Weil Pvt. Ltd. India) anesthesia was used during blood collection to minimize animal discomfort and suffering. For euthanasia, cervical dislocation was implemented. Additionally, animals were humanely euthanized if they were found to be in a moribund state, experiencing obvious pain, or showing signs of severe and prolonged distress during the experimental period. Seventy mice (35 males and 35 females) were randomly assigned to seven groups (Group I—VII). Every Group comprised five animals of each sex for the study. The animals chosen for the study had a body weight variation on the day of randomization that did not exceed 20%, which was around 18–20 g for each group and sex. The randomization process was carried out using an excel sheet [22]. All the test compounds and vehicles were partially blinded to the researchers by coding the test compound. The medicines and their potencies were decoded during the phase of report drafting [22].

## Dose selection

Recent evidence from 28-day oral toxicity studies of *Camphora* [23] and *Rhus tox* [24] shows that administering 20µL/100g orally in 6C, 30C, and 200C potencies is safe for repeated use. Our laboratory's unpublished data from a similar 28-day oral toxicity study on BA in 6C, 30C, and 200C potencies confirm that a 20µL/100g dose volume is safe for repeated use in rats. Janhavi et al. (2022) developed a virtual calculator (DoseCal) for converting doses between different animal species based on the Km (Correction) factor, body weight, and body surface area, following the FDA guidelines of 2005 [25, 26]. Using the DoseCal calculator, the 20µL/100g dose volume for rats converts to 40µL/100g for mice, which was used in the current study.

## Preparation of the BA potencies and vehicle

The potencies of BA were prepared using a standardized procedure in accordance with the Homoeopathic Pharmacopoeia of India (HPI). The dried powder of BA roots were carefully macerated in a solution of 80.47% v/v alcohol, to obtain high-quality mother tincture. One ml of mother tincture was diluted in ninety-nine ml of ethyl alcohol (HPI grade) and succeeded ten times in getting Potency 1. The subsequent potencies were created by diluting one ml of potency 1 with ninety-nine ml of ethyl alcohol and giving ten jerks. Further, this procedure was repeated to obtain the potentized dilutions of 6C, 30C, and 200C [27, 28].

## Preparation of drug and vehicle stock solution

Stock solutions of BA and vehicle (Dispensing alcohol) were prepared by diluting 1 ml of potentized dilution (6C, 30C, and 200C) with 9 ml of double distilled water (1:9 ratio) for administration to experimental animals [29] and all the solutions were blinded until the statistical analysis. Animals in all the groups were administered the respective interventions at a dose volume of 40 µL/100 g, p.o.

## Investigational procedure

The animals were segregated into seven groups, involving 5 males and 5 females each. Then collectively, Group I received distilled water (D/W) (40 µL/100 g, p.o.) and served as normal control; Group II received D/W (40 µL/100 g, p.o.) and maintained as disease control or CPM control. Group III received dispensing alcohol (DA) (40 µL/100 g, p.o.) and served as vehicle control. Group IV served as standard control and received Levamisole (40 mg/kg, p.o.)

dissolved in distilled water. Groups V, VI, and VII received the BA (40 μL/100 g, p.o.) in their potencies, i.e., 6C, 30C & 200C respectively. Except for Group I, all other groups received 80 mg/ kg CPM as a potent cytotoxic drug on days 4, 8 and 12 [30] intraperitoneally for immune suppression. The treatment with respective interventions to all the animals in different groups were given once daily for 14 days.

## Parameters of observation

**Determination of RBC, WBC, and hemoglobin (Hb) count.** The blood samples were collected 24 hr after the last dose administration by puncturing the retro-orbital plexus of the animals using heparinised capillary tubes under isoflurane anesthesia (Raman &Weil Pvt. Ltd. India) using small animal anesthesia system (Orchid Scientific Pvt. Ltd. India) into the K2 EDTA vacutainers (Hemo Tube™, India). A veterinary hematology analyzer (Genrui VH50, China) was employed to test the blood samples for enumerating RBC, WBC, and Hb.

**Determination of spleen and thymus indexes.** On the $15^{th}$ day, 24 hours following the completion of the last dose, animals (n = 6) were euthanised using cervical dislocation method and dissection were performed to obtain spleen and thymus from each Group. The obtained spleen and thymus were weighed, and the organ index was calculated using the following formula [29].

$$Spleen\ index\ \left(\frac{mg}{10g}\right) = Spleen\ weight\ (mg)/Body\ weight\ (g)\ X\ 100 \tag{1}$$

$$Thymus\ index\ \left(\frac{mg}{10g}\right) = Thymus\ weight\ (mg)/Body\ weight\ (g)\ X\ 100 \tag{2}$$

**Phagocytic index assay.** The carbon clearance test was performed to evaluate the effect of test medicines on the phagocytic activity of the reticuloendothelial system (RES). The test was carried out as described by Cheng et al. 2005 and Raj & Gothandam, 2015 [30, 31], with minor modifications. On the $14^{th}$ day of the study, mice (n = 4) were weighed and then injected with pre-warmed Indian liquid ink (0.1 mL/10 g B.W.) through the tail vein, following an interval of 2 hr after the last dose administration. Later, the blood was collected at 2- and 10-min interval from retro-orbital sinus. A 25 μL of blood was mixed with 2 mL of 0.1% sodium carbonate solution. The absorbance was recorded at 660 nm keeping the sodium carbonate solution as blank. Later, mice were euthanized, to excise the spleen, and liver and weighed immediately. The carbon clearance rate (κ) and the phagocytic index (α) were determined using the following formulas:

$$Rate\ of\ carbon\ clearance\ (\kappa) = (\log OD1 - \log OD2)/T2 - T1 \tag{3}$$

$$Rate\ of\ phagocytic\ index\ (\alpha) = 3\sqrt{\kappa}X\ Body\ Weight/(Liver\ weight + Spleen\ weight) \tag{4}$$

**Determination of IL-2, TNF-α, IFN-γ, and IL-4 in serum, and spleen.** Mouse-specific ELISA kits (Elabscience, USA) were used in carrying out tests for the determination of cytokine levels in the serum and spleen of the test animals (n = 6). During the same, the manufacturer's protocol was strictly followed. The serum was obtained from the collected blood. For the other part of the experiment, the protocol explained by Han et al. 2019, with minor modifications was followed [32]. After perfusion, a portion of the spleen was precisely weighed and homogenized in an extraction buffer, with the temperature consistently maintained at 4˚C.

**Table 1. Primer sequence for rtPCR.**

| Gene | Sequence 5' - 3' | Size bp |
|---|---|---|
| IL-2 | F-GCAGCTGTTGATGGACCTAC | 20 |
| | R-TCCACCACAGTTGCTGACTC | 20 |
| IL-4 | F-TCGGCATTTTGAACGAGGTC | 20 |
| | R-GAAAAGCCCGAAAGAGTCTC | 20 |
| TNF-α | F-ATGAGCACAGAAAGCATGATC | 21 |
| | R-TACAGGCTTGTCACTGGAATT | 21 |
| IFN-γ | F-TGAGCAGAGCTCTTGTGGTC | 20 |
| | R-CGTTCCTCCTTGTGGCCTAA | 20 |
| GAPDH | F-5GTGGAGTCTACTGGTGTCTTC3 | 21 |
| | R-5'GTGCAGGAGGCATTGCTTACA3' | 21 |

F- forward; R- Reverse.

The homogenate was then cold-centrifuged at 3500 rpm for 15–20 minutes. The supernatant was carefully pipetted out, and subsequent analysis was conducted using an ELISA reader (Bio Rad, USA) following manufacturer protocol [29].

**Determination of the mRNA expression levels IL-2, TNF-α, IFN-γ, and IL-4 in the spleen.** After 24 hours of last dose administration, i.e., on the 15th day, mice (n = 6) from each group were sacrificed for spleen samples. Later, the spleen samples were weighed and transferred into a pre-chilled microcentrifuge tube and placed on an ice box to avoid RNA degradation. Spleen samples were stored at -20°C for further processing. A spleen tissue sample of ~100 mg was used for RNA extraction. RNA extraction was performed in accordance with the manufacturer's guidelines using the Aurum total RNA mini kit (Bio-Rad, USA). The quantification and verification of RNA purity were conducted using the Spectrophotometer (Thermo Multiskan SkyHigh, USA). cDNA was synthesized from mRNA in accordance with the manufacturer's guidelines using the iScript cDNA synthesis kit (Bio-Rad, USA). cDNA was further diluted 10 times and stored for RT-PCR. The mRNA expression levels of cytokines (IL-2. IL-4, TNF-α, IFN-γ, and GAPDH) were determined by real-time quantitative polymerase chain reaction (qPCR). GAPDH, a housekeeping gene was used as an internal control for cytokines expression to account for variations in mRNA loading. The cytokine's (IL-2. IL-4, TNF-α, IFN-γ, and GAPDH) primers used were detailed in Table 1. The iTaq Universal SYBR Green Supermix kit (Bio-Rad, USA) was used for the real-time PCR reaction. Real-time PCR was performed in a Real-Time fluorescent quantitative PCR detection system (Bio-Rad CFX96, USA). The thermal cycle condition used for reverse transcription was as follows: predenaturation program (3 min at 95°C), amplification and quantification program (5 s at 95°C, 20 s at 59.6°C) with 40 cycles, and melting curve program (0.05 s at 65°C, 5 s at 95°C). The relative quantification of objective genes was calculated according to the $2^{-\Delta\Delta Ct}$ method [33].

**Histopathological analysis of the spleen.** After collection of the whole spleen, it was washed in sterile PBS (pH 7.3 ± 0.1), and without much delay, it was kept in buffered formalin solution for 24 hours. After the specified duration, the spleen was trimmed into 3–5 mm thick pieces for final sectioning. After proper fixation, the tissue was processed in increasing concentrations of alcohol, and completely dehydrated tissue was processed in xylene before it was embedded in liquid paraffin to prepare paraffin-tissue blocks. Using a semi-auto rotary microtome (CIM 18, Biocraft, India), specimen slides with 3–5 μm thick tissue sections were prepared. These slides were stained with Haematoxylin and Eosin (H&E) to analyze the histoarchitecture of the spleen [34]. The images were captured under the magnification of 50X and 200X using a Light microscope (BX43, Evident, Japan).

## Statistical analysis

The data are presented as the means ± SEM. One-way ANOVA and Tukey's multiple comparison tests were conducted to compare multiple groups. All statistical analysis was conducted using the SPSS 21 software. Statistical significance was determined based on values of $p < 0.05$, 0.01, and 0.001.

## Results

### Effect of BA on haematological parameters

Initially, the RBC, WBC and Hb counts were assessed in the CPM induced immune compromised animals and found to be significantly lower ($P < 0.001$) compared to the normal control animals indicating the induction of immunosuppression by CPM. Later, the samples from the BA treated animals in their potencies of 6C, 30C and 200C, showed significantly increased Hb and RBC levels ($p < 0.001$) in comparison with those of CPM and vehicle treated groups. In case of WBC, BA 6C has shown a significant improvement i.e., $P < 0.01$ and $P < 0.05$ in comparison with those in CPM and vehicle treated groups respectively. Whereas, BA 200C showed significance of $P < 0.05$ compared with the CPM and no significance with vehicle treated Group. The above findings indicate that the activity observed is due to the presence of BA but not because of vehicle used, i.e., dispensing alcohol. Levamisole treatment also significantly improved, Hb, RBC ($P < 0.001$) and WBC ($P < 0.01$) levels compared to the CPM control. The results indicate that BA could reverse CPM induced suppression of Hb, RBC, and WBC in mice (Fig 1).

### Effect of BA on spleen and thymus indices

In mice treated with CPM, spleen and thymus indices significantly decreased ($P < 0.05$ and $P < 0.001$, respectively) compared to the normal control. BA 30C ($P < 0.001$) potency significantly improved the spleen index in immunocompromised BALB/c mice. BA 200C potency alone significantly improved ($P < 0.05$) the thymus index, and 6C, 30C improvement were statistically non-significant. BA 30C significantly differed ($P < 0.05$) from the vehicle-treated Group in improving spleen index, in comparison with vehicle treated Group. Levamisole-treated mice showed a significant improvement in the spleen ($P < 0.001$) and thymus ($P < 0.001$) indices than CPM control group. The above data suggested that BA could partially restore CPM induced reduction in spleen and thymus indices (Fig 2A and 2B).

### Effect of BA on the phagocytic index

Similar to the immune organ indices, the phagocytic index of the CPM treated animals was reduced significantly ($P < 0.001$) compared to the normal control group. BA 6C ($P < 0.01$), 30C ($P < 0.001$), 200C ($P < 0.05$) and levamisole ($P < 0.05$) had improved phagocytic function in comparison to CPM control group (Fig 2C).

### Effect of BA on cytokine levels in serum and spleen

CPM treatment caused a significant reduction in the levels of IL-2, IL-4, TNF-$\alpha$, and IFN-$\gamma$ ($P < 0.05/P < 0.001$) in the serum and spleen tissue of mice than normal control, indicating its immunosuppressant effect. The BA 6C treatment has not showed any significant increase in the IL-2 levels of serum and spleen in comparison with CPM control. However, the BA 6C has showed significance ($P < 0.05$) in the serum IL-2 levels with no significance in spleen IL-2 levels in comparison with vehicle control. The BA 30C treatment has showed significant ($P < 0.001$) increase of IL-2 levels in both serum and spleen in comparison with the CPM and

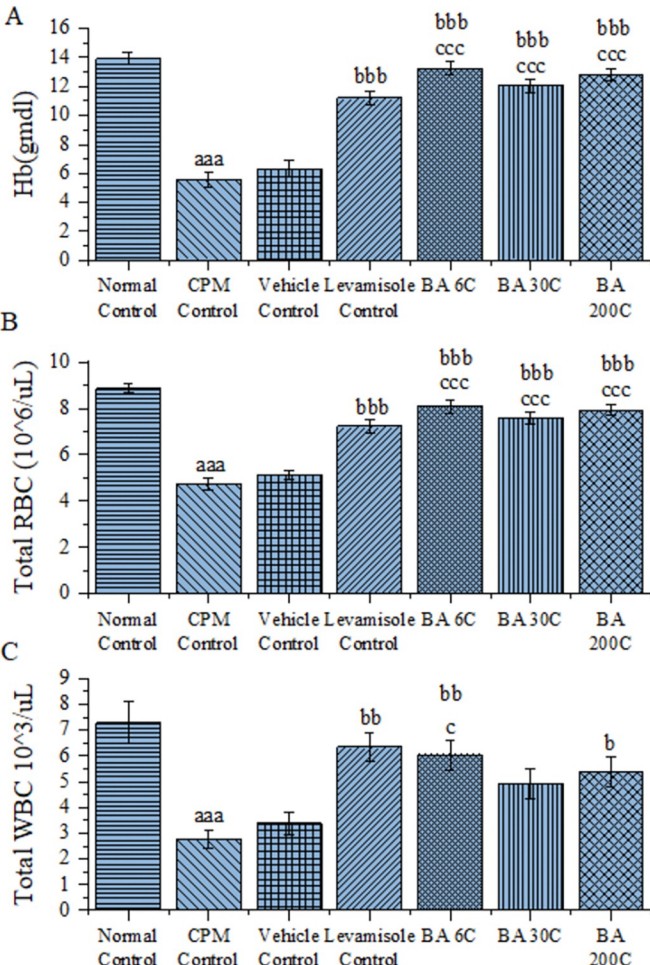

**Fig 1.** Effect of BA on Haematological parameters in CPM-Treated Mice; A- Heamoglobin, B- RBC count, C-WBC count; Data values are expressed as Mean ± SEM (n = 6). Statistical analysis was performed using one-way ANOVA followed by Tukey multiple comparison post hoc test. [aaa]P<0.001 when compared to Normal Control; [b]P<0.05, [bb]P<0.01, [bbb]P<0.001 as compared to the CPM group; [c]P<0.05, [ccc]P<0.001 as compared to the vehicle control group.

vehicle control. BA 200C treatment has demonstrated a significant (P<0.001) rise in IL-2 levels in both serum and spleen, when compared to the vehicle control group. The BA 200C treatment has displayed no significance in serum IL-2 levels with significant (P<0.001) increase in spleen IL-2 levels in comparison with CPM control. The BA 6C, 30C and 200C have displayed a significant (P<0.001, P<0.001 and P<0.05) increase in IL-4 levels in serum and spleen (P<0.001) compared to those in CPM and vehicle control groups.

The BA 6C and BA 200C did not demonstrate any significant rise in serum TNF-α levels. However, the BA 30C exhibited a significant (P<0.001) increase in serum TNF-α levels when compared to the CPM and vehicle control groups. In regards to the spleen TNF-α levels, it is worth noting that BA 6C did not demonstrate any significant findings. However, both BA 30C and 200C exhibited a significant increase (P<0.001) when compared to the CPM and vehicle control groups.

The BA 6C and 200C has showed significant (P<0.001) increase in the serum IFN-γ levels whereas BA 30C has not showed any significance in comparison with CPM and vehicle control. Coming to the spleen IFN-γ levels, BA 6C, 30 C and 200C has showed no significance in

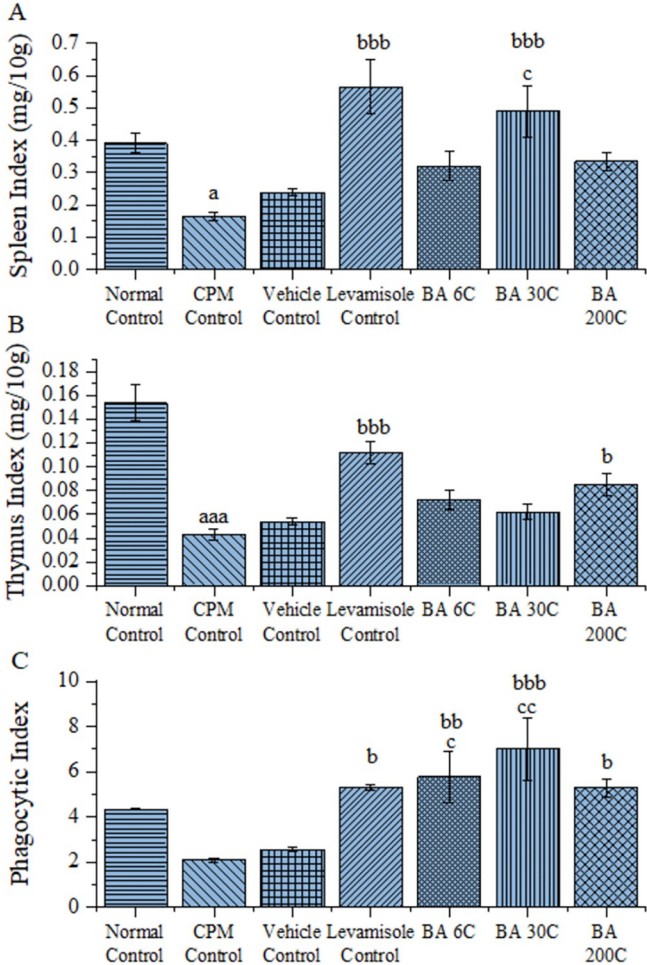

**Fig 2.** Effect of BA on Immune Indices in CPM-Treated Mice; A- Spleen index (n = 6), B- Thymus Index (n = 6), C-Phagocytic index (n = 4); Data values are expressed as Mean ± SEM. Statistical analysis was performed using one-way ANOVA followed by Tukey multiple comparison post hoc test. [a]P<0.05, [aaa]P<0.001 when compared to Normal Control; [b]P<0.05, [bb]P<0.01, [bbb]P<0.001 as compared to the CPM group; [c]P<0.05, [cc]P<0.01 as compared to the vehicle control group.

comparison with CPM and vehicle control. For the positive control (Levamisole), it significantly enhanced (P<0.001) the secretions of serum IL-2, IL-4, IFN-γ and splenic levels of IL-4, TNF-α, and IFN-γ when compared to the CPM control group. These results indicate that BA could enhance immune activity by improving cytokines production in serum and splenocytes of immunosuppressed mice (Fig 3).

### Effect of BA on mRNA expression levels of the cytokines (IL-2, IL-4, TNF-α, and IFN-γ) in spleen

To further confirm the splenic immune enhancement by BA, qRT-PCR was performed to examine the induction of transcriptional upregulation of cytokines in the spleen. As shown in Fig 4, BA significantly increased (P<0.001) the mRNA levels of IL-2, IL-4, TNF-α, and IFN-γ while CPM-treated mice produced substantially lower (P<0.001) mRNA of levels of the above cytokines. Levamisole treatment also significantly upregulated (P<0.001) the mRNA expression of IL-2, IL-4, TNF-α, and IFN-γ genes compared to CPM-treated mice. The mRNA

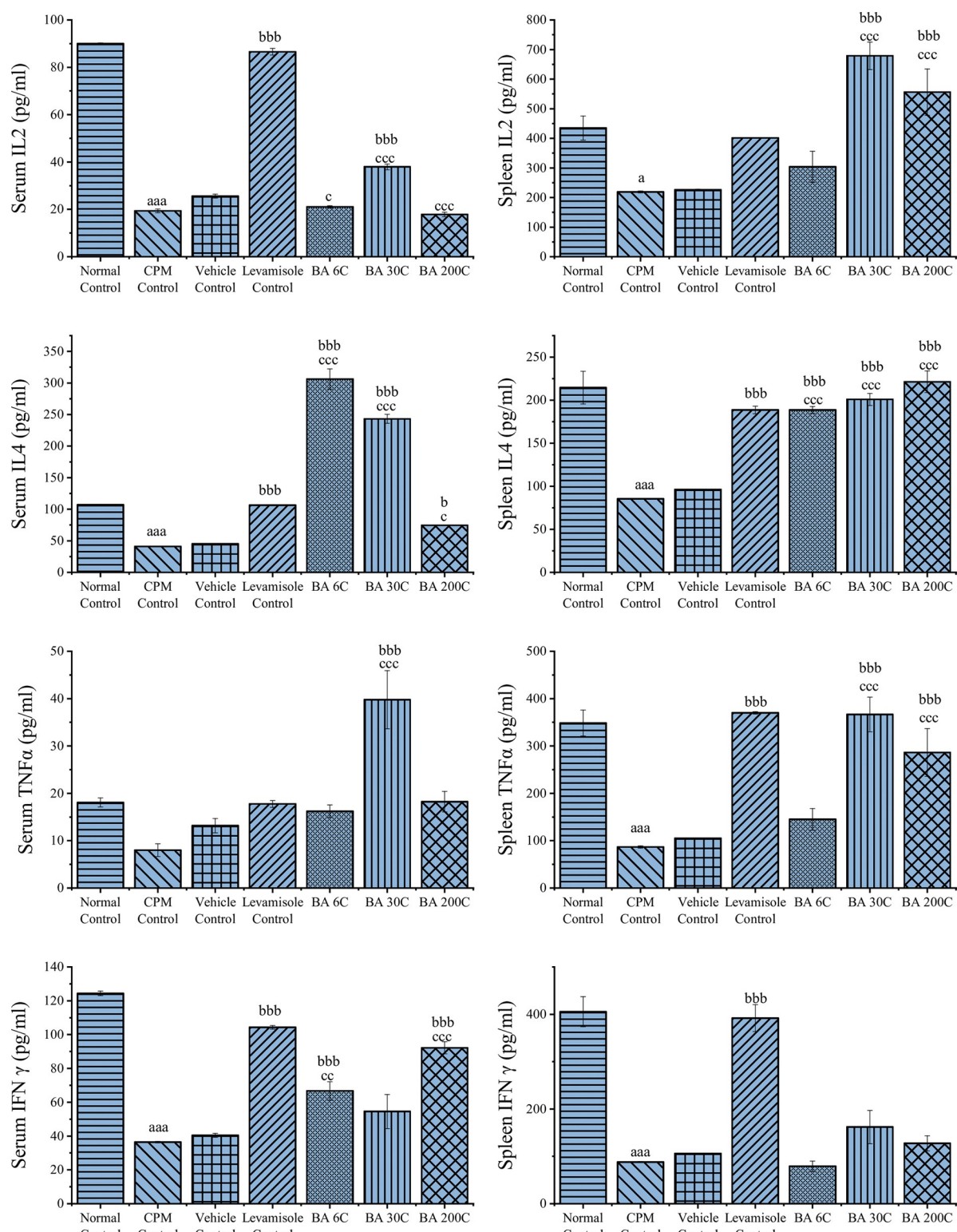

**Fig 3. Effect of BA on serum and spleen cytokine levels in CPM-Treated mice; data values are expressed as mean ± SEM (n = 6).** Statistical analysis was performed using one-way ANOVA followed by Tukey multiple comparison post hoc test. [aaa]$P<0.001$ when compared to Normal Control; [b]$P<0.05$, [bb]$P<0.01$, [bbb]$P<0.001$ as compared to the CPM group; [c]$P<0.05$, [cc]$P<0.01$, [ccc]$P<0.001$ as compared to the vehicle control group.

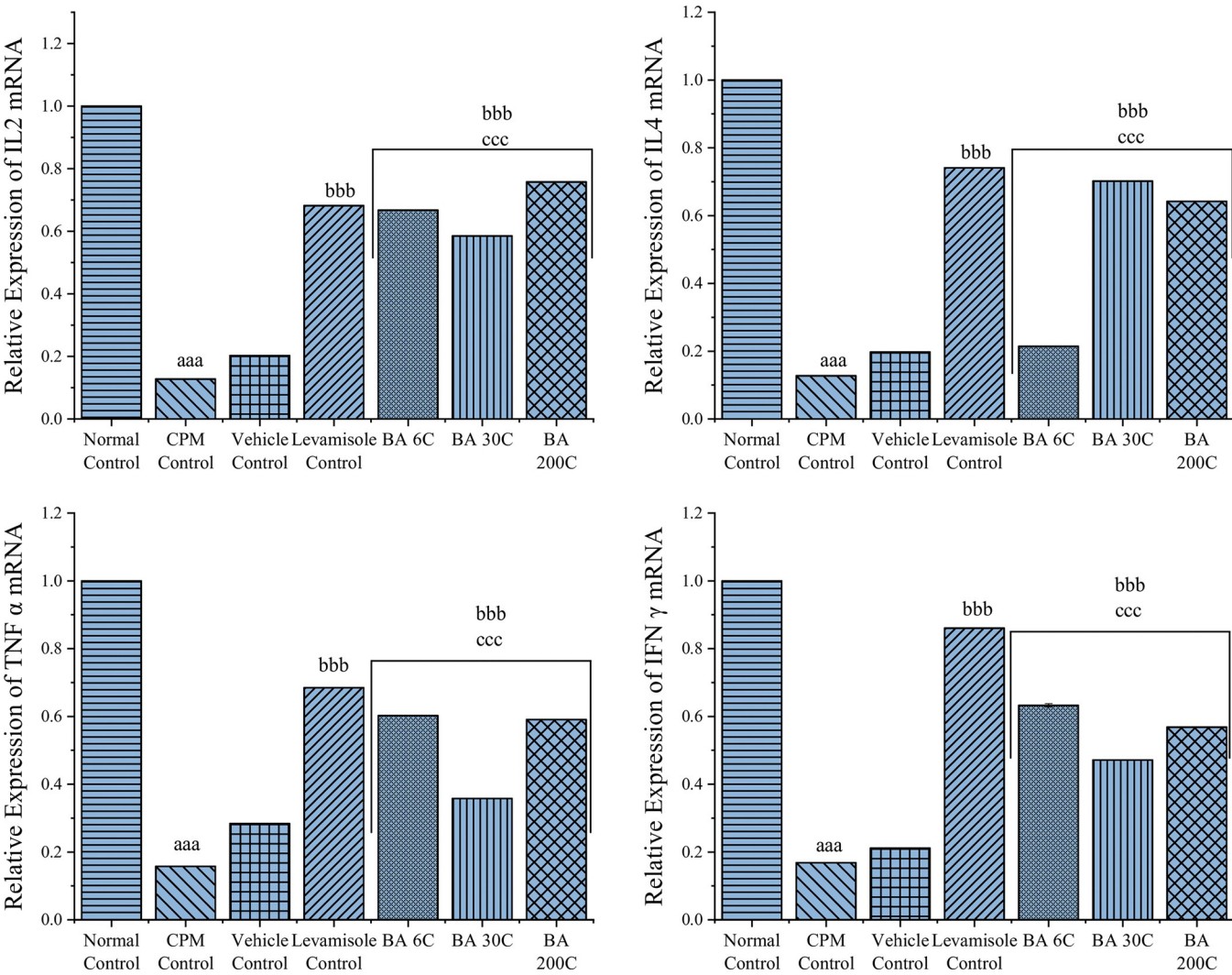

**Fig 4. Effect of BA on mRNA expression levels of the cytokines in the spleen of CPM-treated mice; data values are expressed as mean ± SEM.** n = 6 in each group. Statistical analysis was performed using one-way ANOVA followed by Tukey multiple comparison post hoc tests. [aaa]P<0.001 when compared to Normal Control; [b]P<0.05, [bb]P<0.01, [bbb]P<0.001 as compared to the CPM group; [c]P<0.05, [cc]P<0.01, [ccc]P<0.001 as compared to the vehicle control group.

expression of the above cytokines was significantly different between BA and vehicle-treated groups, further confirming that BA was responsible for reversing the downregulation of mRNA expressions of cytokines induced by CPM in mice.

## Effect of BA on histology of spleen

The histology of spleen sections from normal mice stained with H&E revealed distinct regions of white pulp and red pulp, with the former containing lymphoid tissue and the latter containing splenic cords and venous sinuses, as shown in Fig 5. However, treatment with CPM resulted in a depletion or disruption of the white pulp, with a reduction in germinal centre formation and lymphocyte populations. This was accompanied by an increase in cellularity in the red pulp due to infiltration by immune cells. Additionally, alterations in the structural integrity of the spleen, with fibrosis in the tissue, were observed. The histoarchitecture of the spleen in immunocompromised mice was not protected by alcohol treatment and instead led to white pulp degeneration

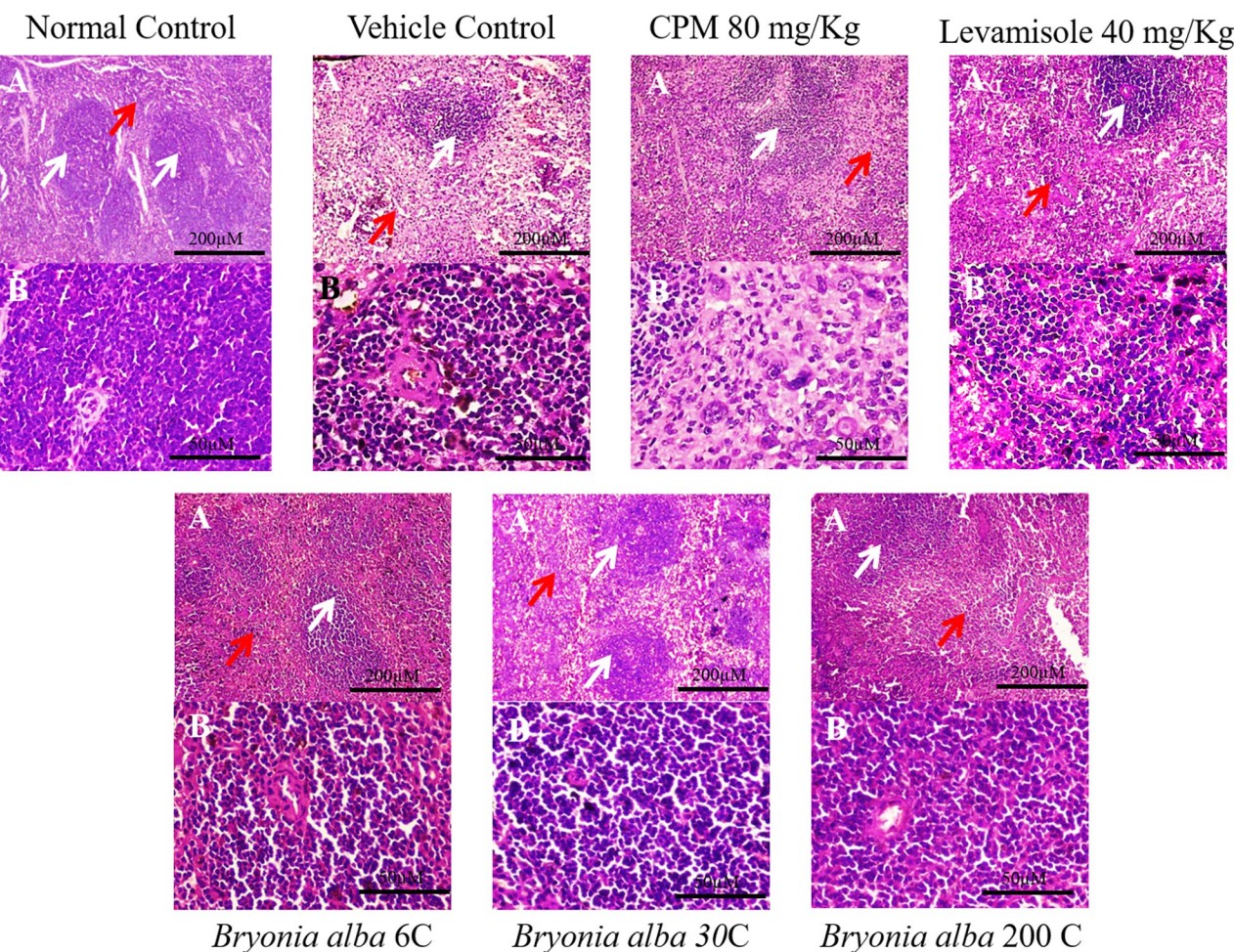

**Fig 5.** Effect of BA on histology of HE stained spleen sections in CPM-treated mice; Plate A with 50X magnification focusing both white and red pulp; Plate B with 200 X magnification focusing white pulp containing lymphocytes; Red arrow indicates red pulp and white arrow indicates white pulp.

and lymphocyte infiltration into the red pulp. On the other hand, treatment with BA in its 6C, 30C, and 200C potencies showed a significant improvement in the histology of the spleen. The arrangement of the red and white pulp improved, indicating the protection from CPM induced damage. Similar protective effect was observed in case of mice treated with levamisole.

## Discussion

There is limited scientific evidence to support the use of BA as an immunomodulatory agent. Some studies have suggested that BA may have anti-inflammatory properties, which could potentially help to regulate the immune system. In this scientific study, the focus was on the potential of BA ultra-high dilutions 6C, 30C & 200C to stimulate the immune system in CPM induced immune suppressed BALB/c mice. CPM is an effective chemotherapeutic drug used in the treatment of various tumors, but it is also known to have a range of negative side effects, including immunosuppression, myelosuppression, and oxidative stress [35]. The immunosuppression resulting from CPM can leave the body vulnerable to infection and other external pathogens. Furthermore, CPM challenge can cause damage and weight loss in the spleen and thymus, which, as representative immune organs, play an essential role in maintaining the body's immune homeostasis [36].

To induce immunosuppression, mice from all groups aside from the normal control group were intraperitoneally given CPM at a dose of 80 mg/kg on days 4, 8, and 12 of the study. Our results indicate that CPM significantly reduced the RBC, WBC, and H.B. levels along with the spleen and thymus indices in comparison with normal control group. The concentrations of IL-2, IL-4, TNF-α, and IFN-γ in the serum and spleen were also reduced by CPM. In addition, CPM mice showed a marked decline in splenic mRNA expression of IL-2, IL-4, TNF-α, and IFN-γ. According to earlier studies and the results presented above, an immunosuppressive mice model using CPM was successful [37, 38].

The normal ranges of white blood cells vary from animal to animal. A higher or lower than normal count could be indicative of a number of illnesses, such as rheumatoid arthritis, allergies, bone marrow deficiencies, specific viral infections, or severe bacterial infections [39]. CPM administration resulted in a significant reduction of RBC, WBC and Hb levels in mice. BA (6C, 30C, and 200C) treatment significantly improved the decline of RBC and Hb levels induced by CPM in mice. BA at 6C and 200C also significantly improved the WBC count in CPM treated mice. The results indicate that BA treatment could enhance the immunity in mice by improving RBC, WBC & Hb levels reduced by CPM.

The body's immune system includes the thymus and spleen, two crucial organs. Reduced size and weight of the spleen and thymus in pathological conditions indicate a decline in immune response [40]. In this study, BA treatment partially improved spleen and thymus indices compared to the CPM control without any statistical significance, suggesting that BA may act to prevent the immune organ atrophy brought on by CPM.

The macrophage function was assessed by carbon clearance test. When serum protein-bound particles of Indian ink containing carbon injected directly into the systemic circulation, macrophages in the liver and spleen intensely swallowed the particles. The rate at which macrophages removed carbon from the blood was determined by an exponential equation, and phagocytic activity was positively correlated with the rate at which carbon particles were removed [41]. Our study implies that the macrophage system's phagocytic activity was reduced by CPM therapy, which concurs with earlier studies [42, 43]. BA treatment had insignificant effects on the phagocytic activity of macrophages compared with the CPM control.

The impact of BA on the synthesis of IL-2, IL-4, TNF-α, and IFN-γ was assessed because cytokines are crucial for triggering an immune response. Numerous cellular and animal studies also demonstrated that enhancing cytokine synthesis enhance immune function in immunodeficient mice [39, 42, 44, 45]. Cytokines function as a signalling molecules among cells that control the immune response, mediate inflammatory responses, participate in immune cell differentiation, and promote hematopoietic function [46]. A cell growth factor called IL-2 encourages cell division and proliferation. Interferon production and inflammatory or autoimmune reactions can both be facilitated by IL-2. TNF-α is a crucial component of the host defence system and kills tumour cells. Additionally, TNF-α can promote the expression of a number of inflammatory and immune mediators. The biological activities of IFN-γ are primarily to activate macrophages, inhibit viral replication, and induce the expression of MHC molecules. Th2, mast, and natural killer (NK) cells secrete IL-4, which interacts with T, B, macrophages, and mast cells to activate and differentiate B cells, increase class II MHC expression, and induce and grow T cells and mast cells, respectively [43, 47]. Our study results show that BA had an excellent regulatory effect on cytokines production in CPM induced immunosuppressed mice. BA at 30C raised the concentrations of IL-2, TNF-α, and IL-4 in mouse serum, 30C and 200C potencies raised the levels of the IL-2, TNF-α, and IL-4 in the splenic cytokines indicating the effect of BA on splenocytes also. This finding supports BA's ability to balance the body's immune system and lessen immunosuppression symptoms.

qRT-PCR was performed on spleen cytokines, such as IL-2, IL-4, TNF-α, and IFN-γ, to further support the immune enhancement of BA in the spleen from a CPM-induced immunosuppressed state. The results revealed a significant increase in the mRNA levels of IL-2, IL-4, TNF-α, and IFN-γ in the spleen of BA treated mice, compared to the CPM group, which produced lower levels of these cytokines. Through the current investigation, it was revealed that BA had significantly improved the cytokine production in mice undergoing immunosuppression with CPM, displaying moderate yet noteworthy immunostimulant potency.

The histological stained sections of spleen exhibited interesting data that are in good agreement with the existing piece of evidences. The normal group displayed intact white and red pulp proving that the spleen is in good health. The CPM treated group displayed disruption, degeneration and atrophy in the spleen, indicating its potential harm [37, 38]. However, the BA administration demonstrated a protective impact by exhibiting intact white and red pulp with well-defined borders and a higher concentration of lymphocytes in the spleen. Levamisole treatment also displayed similar positive effects.

The study emphasizes the therapeutic benefits of BA in safeguarding the spleen against damage caused by chemotherapy. The findings also indicate that alcohol, when used as a vehicle for BA, does not impact its immunostimulant properties, as demonstrated by the lack of improvement in CPM-treated mice. Additionally, the study found that the immunoenhancement activity of BA potencies (6C, 30C, and 200C) is comparable to that of the standard immunostimulant control, Levamisole.

The study's limitations encompass the utilization of a mice model to simulate chemotherapy-induced immunosuppression through CPM administration. While this model is widely employed, it might not entirely replicate the intricate immunological responses observed in human chemotherapy patients. Despite the assessment of cytokine levels and immune cell markers, a deeper exploration of BA intricate immunomodulatory effects, including the investigation of molecular signalling pathways and cellular interactions, could provide valuable insights into its mode of action. Despite these constraints, the research underscores BA potential therapeutic benefits in mitigating chemotherapy-induced immune system impairment.

## Conclusion

The present study proves, for the first time, that extremely high dilutions of BA 6C, 30C, and 200C potencies could lessen the adverse effects of CPM on immune function in immune compromised mice. Based on the results, though the precise underlying mechanism of BA is unknown, we conclude that BA's ability to increase RBC and WBC levels, spleen index, IL-2, IL-4, TNF-α, and IFN-γ cytokine levels in serum & spleen are responsible for the immune stimulant activity it imparts. Above all, our research may offer a mechanistic rationale for using BA -6C, 30C, and 200C as effective adjuvant immunostimulant therapy or a different strategy to lessen CPM induced immunosuppression. Moreover, the study's methodology and approach could serve as a blueprint for exploring BA immunomodulatory potential in diverse immune-related disorders beyond chemotherapy-induced immunosuppression. These results also provide a solid foundation for future investigations into the immunomodulatory potential of BA clinically, which could have important implications for the development of homoeopathic medicines as immunomodulatory agents.

## Supporting information

**S1 File. Supporting data corresponding to the Figs 1–4 in the manuscript.**
(XLSX)

## Acknowledgments

The authors would like to thank Arti Soren, Assistant Director, Dr. Anjali Chatterji Regional Research Institute for Homoeopathy, Kolkata for her guidance and administrative support. The authors would also like to thank N. P. Dongre, Pathology department, for his technical support.

## Author Contributions

**Conceptualization:** Vara Prasad Saka, Godlaveti Vijay Narasimha Kumar, Pankaj Gupta.

**Formal analysis:** Vara Prasad Saka, Abanti Goswami.

**Funding acquisition:** Subhash Kaushik.

**Investigation:** Vara Prasad Saka, Abanti Goswami, Pankaj Gupta.

**Methodology:** Godlaveti Vijay Narasimha Kumar.

**Project administration:** Digvijay Verma.

**Supervision:** Godlaveti Vijay Narasimha Kumar, Pankaj Gupta.

**Visualization:** Bharat Kumar Reddy Sanapalli.

**Writing – review & editing:** Bharat Kumar Reddy Sanapalli, Subhash Kaushik.

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
