## [Decision Letter · Decision Letter 0]

24 Jun 2024

PONE-D-24-16624Modulatory Action of Bryonia alba on the Immune System in Cyclophosphamide Induced Immunosuppression in BALB/c MicePLOS ONE

Dear Dr. Kumar,

Thank you for submitting your manuscript to PLOS ONE. After careful consideration, we feel that it has merit but does not fully meet PLOS ONE’s publication criteria as it currently stands. Therefore, we invite you to submit a revised version of the manuscript that addresses the points raised during the review process. Kindly take into account the reviewers' comments to enhance the manuscript's quality. Please submit your revised manuscript by Aug 08 2024 11:59PM. If you will need more time than this to complete your revisions, please reply to this message or contact the journal office at plosone@plos.org. Please include the following items when submitting your revised manuscript:A rebuttal letter that responds to each point raised by the academic editor and reviewer(s). You should upload this letter as a separate file labeled 'Response to Reviewers'.A marked-up copy of your manuscript that highlights changes made to the original version. You should upload this as a separate file labeled 'Revised Manuscript with Track Changes'.An unmarked version of your revised paper without tracked changes. You should upload this as a separate file labeled 'Manuscript'.If applicable, we recommend that you deposit your laboratory protocols in protocols.io to enhance the reproducibility of your results. Protocols.io assigns your protocol its own identifier (DOI) so that it can be cited independently in the future. For instructions see: https://journals.plos.org/plosone/s/submission-guidelines#loc-laboratory-protocols. Additionally, PLOS ONE offers an option for publishing peer-reviewed Lab Protocol articles, which describe protocols hosted on protocols.io. Read more information on sharing protocols at https://plos.org/protocols?utm_medium=editorial-email&utm_source=authorletters&utm_campaign=protocols.

We look forward to receiving your revised manuscript.

Kind regards,

Wesley Lyeverton Correia Ribeiro, Ph.D.

Academic Editor

PLOS ONE

2. To comply with PLOS ONE submissions requirements, in your Methods section, please provide additional information regarding the experiments involving animals and ensure you have included details on (1) methods of sacrifice, and (2) efforts to alleviate suffering.

 [Central Council for Research in Homoeopathy, India sponsored the study under the Intramural Research Scheme (4-13/2019-20/CCRH/Tech/DS/Kolkata/DSU/3255 dated 18-03-2021).].  

5. We note that your Data Availability Statement is currently as follows: [All relevant data are within the manuscript and its Supporting Information files.]

6. We note that Figure 5 in your submission contain copyrighted images. All PLOS content is published under the Creative Commons Attribution License (CC BY 4.0), which means that the manuscript, images, and Supporting Information files will be freely available online, and any third party is permitted to access, download, copy, distribute, and use these materials in any way, even commercially, with proper attribution. For more information, see our copyright guidelines: http://journals.plos.org/plosone/s/licenses-and-copyright.

a. You may seek permission from the original copyright holder of Figure 5 to publish the content specifically under the CC BY 4.0 license. 

7. Please include a caption for figure 5.

Reviewers' comments:

Reviewer's Responses to Questions

**Comments to the Author**

1. Is the manuscript technically sound, and do the data support the conclusions?

Reviewer #1: Yes

Reviewer #2: Yes

2. Has the statistical analysis been performed appropriately and rigorously? 

Reviewer #1: Yes

Reviewer #2: Yes

3. Have the authors made all data underlying the findings in their manuscript fully available?

Reviewer #1: Yes

Reviewer #2: Yes

4. Is the manuscript presented in an intelligible fashion and written in standard English?

Reviewer #1: Yes

Reviewer #2: Yes

5. Review Comments to the Author

Reviewer #1: I suggest that the author pays attention to the bibliographical references in the introduction. I consider the references used to be somewhat outdated for the publication, I try citations that have been published for more than 14 years.

Reviewer #2: Some points to improve

In the introduction, some improvements can be implemented to strengthen the scientific basis and clarity of the text. Initially, it would be beneficial to update the references used to support claims about homeopathy and Bryonia alba, incorporating more recent and robust studies to increase the credibility of the information presented. Furthermore, it is important to detail the evidence mentioned in the introduction. Although several studies on the use of BA are cited, there is a lack of specifications on study design, sample sizes, and specific results. Including more details could strengthen the claims and provide a clearer view of the current scientific landscape.

To improve the accessibility of the text, it could be useful to briefly explain some technical terms, such as "immunosuppression" and "homeopathic potencies", making it more understandable for non-specialist readers. In addition, small grammatical and concordance errors must be reviewed to ensure the fluidity and accuracy of the text.

Regarding the mechanisms of action of BA, the introduction mentions its immunomodulatory and antioxidant characteristics, but does not offer a detailed explanation of the possible underlying biological mechanisms. A deeper analysis of these mechanisms would not only increase understanding but also reinforce the credibility of the conclusions presented. The rationale for the study can also be expanded. While the introduction states that the immunostimulating properties of BA in situations of immunosuppression have not been adequately explored, it would be useful to explain in more detail why this gap in knowledge is relevant and what the potential clinical impacts of this research are. Finally, discussing the challenges inherent in researching homeopathic treatments, such as the variability of results and the difficulty of replication, would provide a more balanced and critical view of the study, offering a more complete perspective for readers.

In the methodology, some improvements can be suggested to strengthen the robustness of the study. First, it would be beneficial to supplement the references used (e.g., Singh et al., 2019; Jahnavi et al., 2019) with more recent studies or updated literature reviews that solidify the scientific basis for the doses and procedures employed. Additionally, the list of reagents and chemicals could be expanded to include all critical materials used in the study, such as ELISA kits, reagents for RNA and cDNA, and the specific equipment used in PCR and microscopy analyses. The justification for the choice of dose, although it mentioned the conversion of the dose from albino rats to albino mice, could be more robust. It would be helpful to provide better reasoning, perhaps with additional references or a more detailed explanation of the conversion process used to determine the final volume (40 μL/100 g). Furthermore, some minor procedural details, such as the exact protocol for retro-orbital blood collection, could be more thoroughly described. This would ensure that all procedures can be replicated accurately, promoting the consistency and reliability of the results obtained.

The discussion recognizes the limitation of using an animal model to simulate chemotherapy-induced immunosuppression, which may not fully replicate human immune responses. This recognition is important as it highlights the need for clinical studies in humans to validate these findings. Research could benefit from further exploration of the molecular mechanisms by which BA exerts its immunomodulatory effects. Investigating cellular signaling pathways and molecular interactions would provide a more complete understanding of BA's modes of action. The discussion addresses the use of alcohol as a vehicle for BA and its lack of impact on the observed immunomodulatory properties, which is relevant to ensure that the beneficial effects are due to the BA and not the vehicle. Comparison with Levamisole, a known immunomodulator, strengthens the results, demonstrating that BA has comparable efficacy, which is encouraging for the validation of BA as a potential therapeutic agent.

The conclusion effectively summarizes the main results, such as increased levels of RBC, WBC, spleen indices and cytokines (IL-2, IL-4, TNF-α, IFN-γ) in serum and spleen, reinforcing the evidence that BA has a relevant immunomodulatory action. While recognizing that the mechanism of action of BA is still unknown, it would be useful to suggest specific methods or approaches to explore these mechanisms in future research, adding value and direction to studies. It is important to reinforce caution when generalizing results from animal models to humans, contextualizing the findings within the limits of preclinical research. The conclusion could include a brief reference to previous studies on immunomodulation, better situating the findings within the broader body of scientific literature. Furthermore, explicitly suggesting clinical trials to validate the effects in humans would be valuable, showing a clear path to practical application of the results.

6. PLOS authors have the option to publish the peer review history of their article (what does this mean?). If published, this will include your full peer review and any attached files.

Reviewer #1: No

Reviewer #2: **Yes: **Jéssica Rabelo Bezerra

---

## [Author Response · Author response to Decision Letter 0]

2 Aug 2024

Responses to Reviewer 1 Comments:

Comment: 1

I suggest that the author pays attention to the bibliographical references in the introduction. I consider the references used to be somewhat outdated for the publication, I try citations that have been published for more than 14 years.

Response: Thank you for your valuable feedback. We acknowledge that some of the references in the introduction are outdated. We have updated the introduction with more recent and relevant studies to support our claims. This includes incorporating studies from the past decade to provide a current perspective on the topic. 

Responses to Reviewer 2 Comments:

Comment: 1

In the introduction, some improvements can be implemented to strengthen the scientific basis and clarity of the text. Initially, it would be beneficial to update the references used to support claims about homeopathy and Bryonia alba, incorporating more recent and robust studies to increase the credibility of the information presented. Furthermore, it is important to detail the evidence mentioned in the introduction. Although several studies on the use of BA are cited, there is a lack of specifications on study design, sample sizes, and specific results. Including more details could strengthen the claims and provide a clearer view of the current scientific landscape.

Response: Thank you for your insightful comments. We have updated the references in the introduction to include more recent and robust studies, which enhances the credibility of our information on homeopathy and Bryonia alba. Additionally, we have provided more detailed descriptions of the cited studies, including their design, sample sizes, and specific results, to enhance the scientific basis and clarity of the text. 

Comment: 2

To improve the accessibility of the text, it could be useful to briefly explain some technical terms,

such as "immunosuppression" and "homeopathic potencies", making it more understandable for non-specialist readers. In addition, small grammatical and concordance errors must be reviewed to ensure the fluidity and accuracy of the text.

Response: We have added brief explanations for technical terms like "immunosuppression" and "homeopathic potencies" to enhance the accessibility of the text for non-specialist readers. Additionally, we have reviewed and corrected small grammatical and concordance errors to ensure better fluidity and accuracy. 

Comment: 3

Regarding the mechanisms of action of BA, the introduction mentions its immunomodulatory and antioxidant characteristics, but does not offer a detailed explanation of the possible underlying biological mechanisms. A deeper analysis of these mechanisms would not only increase understanding but also reinforce the credibility of the conclusions presented. The rationale for the study can also be expanded. While the introduction states that the immunostimulating properties of BA in situations of immunosuppression have not been adequately explored, it would be useful to explain in more detail why this gap in knowledge is relevant and what the potential clinical impacts of this research are. Finally, discussing the challenges inherent in researching homeopathic treatments, such as the variability of results and the difficulty of replication, would provide a more balanced and critical view of the study, offering a more complete perspective for readers.

Response: We agree with your suggestion. In the revised manuscript, we have provided a more detailed explanation of the possible underlying biological mechanisms of BA's immunomodulatory and antioxidant characteristics. Additionally, we have expanded the rationale for the study, explaining the relevance of the knowledge gap and the potential clinical impacts. We have also discussed the challenges inherent in researching homeopathic treatments, such as variability of results and difficulty of replication, to provide a more balanced and critical view of the study. 

Comment: 4

In the methodology, some improvements can be suggested to strengthen the robustness of the study. First, it would be beneficial to supplement the references used (e.g., Singh et al., 2019; Jahnavi et al., 2019) with more recent studies or updated literature reviews that solidify the scientific basis for the doses and procedures employed. Additionally, the list of reagents and chemicals could be expanded to include all critical materials used in the study, such as ELISA kits, reagents for RNA and cDNA, and the specific equipment used in PCR and microscopy analyses. The justification for the choice of dose, although it mentioned the conversion of the dose from albino rats to albino mice, could be more robust. It would be helpful to provide better reasoning, perhaps with additional references or a more detailed explanation of the conversion process used to determine the final volume (40 μL/100 g). Furthermore, some minor procedural details, such as the exact protocol for retro-orbital blood collection, could be more thoroughly described. This would ensure that all procedures can be replicated accurately, promoting the consistency and reliability of the results obtained.

Response: Thank you for your valuable feedback. In the revised manuscript, we have updated the references in the methodology section with more recent studies and literature reviews. We have also expanded the list of reagents and chemicals to include all critical materials used in the study. The justification for the choice of dose has been strengthened with additional references and a more detailed explanation of the conversion process. Furthermore, we have provided a more thorough description of minor procedural details, such as the exact protocol for retro-orbital blood collection, to ensure accurate replication of procedures. 

Comment: 5

The discussion recognizes the limitation of using an animal model to simulate chemotherapy-induced immunosuppression, which may not fully replicate human immune responses. This recognition is important as it highlights the need for clinical studies in humans to validate these findings. Research could benefit from further exploration of the molecular mechanisms by which BA exerts its immunomodulatory effects. Investigating cellular signaling pathways and molecular interactions would provide a more complete understanding of BA's modes of action. The discussion addresses the use of alcohol as a vehicle for BA and its lack of impact on the observed immunomodulatory properties, which is relevant to ensure that the beneficial effects are due to the BA and not the vehicle. Comparison with Levamisole, a known immunomodulator, strengthens the results, demonstrating that BA has comparable efficacy, which is encouraging for the validation of BA as a potential therapeutic agent.

Response: Thank you for your constructive feedback. As discussed in the limitations, we are positively looking forward to extend the work to find out the molecular signaling pathways of BA.

Comment: 6

The conclusion effectively summarizes the main results, such as increased levels of RBC, WBC, spleen indices and cytokines (IL-2, IL-4, TNF-α, IFN-γ) in serum and spleen, reinforcing the evidence that BA has a relevant immunomodulatory action. While recognizing that the mechanism of action of BA is still unknown, it would be useful to suggest specific methods or approaches to explore these mechanisms in future research, adding value and direction to studies. It is important to reinforce caution when generalizing results from animal models to humans, contextualizing the findings within the limits of preclinical research. The conclusion could include a brief reference to previous studies on immunomodulation, better situating the findings within the broader body of scientific literature. Furthermore, explicitly suggesting clinical trials to validate the effects in humans would be valuable, showing a clear path to practical application of the results.

Response: Thank you for your constructive feedback. We had modified the conclusion in the manuscript.

---

## [Editor Report · Decision Letter 1]

15 Aug 2024

Modulatory Action of Bryonia alba on the Immune System in Cyclophosphamide Induced Immunosuppression in BALB/c Mice

PONE-D-24-16624R1

Dear Dr. Kumar,

We’re pleased to inform you that your manuscript has been judged scientifically suitable for publication and will be formally accepted for publication once it meets all outstanding technical requirements.

Kind regards,

Wesley L. C. Ribeiro, Ph.D.

Academic Editor

PLOS ONE

Additional Editor Comments (optional):

Reviewers' comments:

The authors have adequately considered all reviewers' comments.

---

## [Editor Report · Acceptance letter]

27 Aug 2024

PONE-D-24-16624R1 

PLOS ONE

Dear Dr. Kumar, 

I'm pleased to inform you that your manuscript has been deemed suitable for publication in PLOS ONE. Congratulations! Your manuscript is now being handed over to our production team.

Kind regards, 

on behalf of

Dr. Wesley Lyeverton Correia Ribeiro 

Academic Editor

PLOS ONE